# Toll-like Receptor-Mediated Immunomodulation of Th1-Type Response Stimulated by Recombinant Antigen of Type 2 Porcine Reproductive and Respiratory Syndrome Virus (PRRSV-2)

**DOI:** 10.3390/v15030775

**Published:** 2023-03-17

**Authors:** Rika Wahyuningtyas, Mei-Li Wu, Wen-Bin Chung, Hso-Chi Chaung, Ko-Tung Chang

**Affiliations:** 1Research Centre for Animal Biologics, National Pingtung University of Science and Technology, Pingtung 91201, Taiwan; 2Department of Veterinary Medicine, National Pingtung University of Science and Technology, Pingtung 91201, Taiwan; 3Department of Food Science, National Pingtung University of Science and Technology, Pingtung 91201, Taiwan; 4Flow Cytometry Center, Precision Instruments Center, National Pingtung University of Science and Technology, Pingtung 91201, Taiwan; 5Department of Biological Science and Technology, National Pingtung University of Science and Technology, Pingtung 91201, Taiwan

**Keywords:** porcine alveolar macrophages, porcine reproductive and respiratory syndrome virus, M1, M2, Toll-like receptors

## Abstract

PRRSV infects CD163-positive macrophages and skews their polarization toward an M2 phenotype, followed by T-cell inactivation. In our previous study, we found that recombinant protein A1 antigen derived from PRRSV-2 was a potential vaccine or adjuvant for immunization against PRRSV-2 infection due to its ability to repolarize macrophages into M1 subtype, thereby reducing CD163 expression for viral entry and promoting immunomodulation for Th1-type responses, except for stimulating Toll-like receptor (TLR) activation. The aim of our current study was to evaluate the effects of another two recombinant antigens, A3 (ORF6L5) and A4 (NLNsp10L11), for their ability to trigger innate immune responses including TLR activation. We isolated pulmonary alveolar macrophages (PAMs) from 8- to 12-week-old specific pathogen free (SPF) piglets and stimulated them with PRRSV (0.01 MOI and 0.05 MOI) or antigens. We also investigated the T-cell differentiation by immunological synapse activation of PAMs and CD4^+^ T-cells in the cocultured system. To confirm the infection of PRRSV in PAMs, we checked the expression of *TLR3*, *7*, *8*, and *9*. Our results showed that the expression of *TLR3*, *7*, and *9* were significantly upregulated in PAMs by A3 antigen induction, similar to the extent of PRRSV infection. Gene profile results showed that A3 repolarizes macrophages into the M1 subtype potently, in parallel with A1, as indicated by significant upregulation of proinflammatory genes (*TNF-α*, *IL*-*6*, *IL*-*1β* and *IL-12*). Upon immunological synapse activation, A3 potentially differentiated CD4 T cells into Th1 cells, determined by the expression of IL-12 and IFN-γ secretion. On the contrary, antigen A4 promoted regulatory T cell (T-reg) differentiation by significant upregulation of *IL-10* expression. Finally, we concluded that the PRRSV-2 recombinant protein A3 provided better protection against PRRSV infection, suggested by its capability to reeducate immunosuppressive M2 macrophages into proinflammatory M1 cells. As M1 macrophages are prone to be functional antigen-presenting cells (APCs), they can call for TLR activation and Th1-type immune response within the immunological synapse.

## 1. Introduction

PRRSV, or porcine reproductive and respiratory syndrome virus, infects pigs and causes significant economic losses for pig farmers [1,2,3,4]. PRRSV is a single-stranded RNA virus belonging to the family Arteriviridae. The PRRSV genome consists of a positive-sense RNA molecule, which is approximately 15 kb in length [5]. The genome contains at least 10 open reading frames (ORFs), including ORF1a, ORF1b, ORF2a, ORF2b, and ORF3-7, which encode for viral structural and nonstructural proteins (NSPs): NSP1α, NSP1β, and NSP2 to NSP12 [6,7]. PRRSV is transmitted mainly through direct contact with infected pigs or their body fluids, such as nasal secretions, saliva, semen, and feces [8,9]. Indirect transmission can also occur through contaminated fomites, such as clothing and equipment. In addition, the virus can be transmitted vertically from infected sows to their offspring during pregnancy [10,11,12]. Porcine alveolar macrophages (PAMs) are the main target cells for PRRSV replication in the body. This is because the virus has a preference for cells of the monocyte–macrophage lineage and is able to replicate efficiently in PAMs. As a result, PAMs play a crucial role in the immune response to PRRSV infection and are an important focus of vaccine development [13,14]. There are several vaccines available for PRRSV currently, but their efficacy can vary. One of the major limitations of these vaccines is that they may not effectively stimulate the Toll-like receptors (TLRs) of the pig immune system [15]. The activation of TLRs by PRRSV vaccine has been proposed as a potential mechanism to inhibit replication of the virus. TLR stimulation also plays a significant role in the fate of CD4^+^ T helper 1 (Th1) cells during infection or vaccination [16]. To protect against most pathogens, it is necessary to stimulate Th1 immune responses. Upon encountering a virus, Th1 cells become activated and produce cytokines, including interferon-gamma, which trigger other immune cells to attack and remove the virus [17]. Th1 cells also play a role in memory immunity, which helps the body to remember and respond more effectively to a previously encountered virus by promoting the development of long-lived memory B and T cells [18].

PRRSV vaccines are typically designed to induce an immune response by exposing the pig to an avirulent or inactivated form of the virus [19]. However, if the vaccine does not effectively mimic the viral molecules that bind to the TLRs, the vaccine may not be able to effectively activate the immune response, thereby reducing its efficacy. There are several ways to increase the efficacy of vaccines targeting TLRs, such as using the codelivery of costimulatory molecules with the antigens [20]. The costimulatory proteins CD40, CD80 (B7.1), CD86 (B7.2), and CD70 are upregulated on antigen-presenting cells (APCs) after TLR activation [21].

Another significance of TLRs in vaccination is their capability to induce the activation of cross-priming CD8^+^ T cells, a crucial factor in enhancing the efficacy of vaccine responses [15]. Cross-priming is the process by which naïve CD8^+^ T cells are activated by APCs that have taken up antigens from other cells [22]. This process helps the body mount a strong immune response to vaccines that target intracellular pathogens that require cytotoxic lymphocyte (CTL)-mediated immunity. It has been shown to occur through the stimulation of TLR3 and TLR9 [23]. TLR3 stimulation led to cross-priming in a vaccine model using virus-infected cells, while CpG ODNs stimulate B cells and DCs to initiate cross-priming through the TLR9 pathway [24,25,26].

However, it is important to note that development of a PRRSV vaccine is complicated. There are multiple genetic strains of the virus, and it can adapt to new environment and immune pressure, which further complicate the situation. Thus, developing a PRRSV vaccine can be a challenging task. In our previous study, we found that a recombinant protein derived from PRRSV-2 named A1 antigen, which consists of the complete sequence of ORF5 and a partial sequence of ORF6, as well as T-cell epitopes, can stimulate the repolarization of M2 PAMs to M1 and activate the Th1 response [27]. However, there was a lack of response of TLRs to this A1 antigen. Based on this bias, we hypothesized that changes in the amino acid composition of the spike protein may improve antigen recognition, specifically for certain TLRs, by APCs. Our recently developed antigen, A3, which was also produced from PRRSV-2, shares similar components with A1, but has a partial modification of ORF5 and ORF6 sequences. It possesses the remarkable ability not only to repolarize M1 PAMs but also boost the expression of *TLR3*, *TLR7*, and *TLR9* in PAMs, resulting in an immunological synapse activation of the Th1 cells. In contrast, the other antigen (A4), composed of ORF7, NSP10, and NSP11, does not possess such abilities. Our findings indicated that A3 is providing a more promising option for a vaccine as it not only can reduce viral entry by inducing M1 macrophage repolarization but also trigger TLR activation and Th1 immune response of macrophages as a well-functioning APC other than just a host cell of PRRSV.

## 2. Materials and Methods

### 2.1. Ethics Statement

The Institutional Animal Care and Use Committee at the National Pingtung University of Science and Technology in Taiwan granted approval for the acquisition of lung tissue and the euthanizing of pigs (NPUST-106-053).

### 2.2. Pigs and Inoculations

This research was conducted using specific pathogen-free (SPF) piglets that were around eight to eleven weeks old and weighed nine to twelve kilograms. These piglets were raised in a positively pressurized room at the National Pingtung University of Science and Technology Animal Diagnostic Center.

### 2.3. Construction of the Recombinant Protein Antigen

Antigen A3 (ORF6L5) was constructed from the complete sequence of ORF5 combined with the partial sequence of ORF6 (Appendix A). Antigen A4 (NLNsp10L11) was constructed by the combination of ORF7 with NSP10 and NSP11 (Appendix A). The recombinant antigens were specifically engineered to include BamHI and EcoRI sites to facilitate efficient cloning and expression of the target protein. These sequences were then expressed using the baculovirus expression system.

### 2.4. Collecting Porcine Alveolar Macrophages (PAMs)

The pigs were put down through exsanguination, and to prevent the lungs from fully collapsing, the trachea was secured. Afterwards, the heart and lungs were extracted from the chest, and alveolar macrophages were gathered from the fresh lungs in a sterile manner. To obtain the alveolar macrophages, the lungs were rinsed with phosphate-buffered saline a few times through the trachea, and the resulting solution containing the alveolar macrophages was centrifuged for 10 min. The collected alveolar macrophages were placed in 12-well plates and maintained in a complete RPMI-1640 (Corning, Manassas, VA, USA) medium containing 10% fetal bovine serum at 37 °C in a humidified 5% CO_2_ atmosphere.

### 2.5. PRRSV-Infected PAMs

PAMs were placed in 24-well plates and allowed to grow until they formed a confluent monolayer. At that point, the PAMs were infected with PRRSV in multiples of infection (MOI) of 0.01 and 0.05. The PRRSV inoculum in 200 μL of Optipro serum-free medium was added to each well. After one hour of incubation at 37 °C in a 5% CO_2_ atmosphere, the medium was removed from the wells, and the cells were washed with phosphate buffered saline (PBS). Then, 500 μL of PAM culture medium was added to each well. The cell culture supernatants were collected 48 h after infection.

### 2.6. PAMs Received Antigen Stimulations

To conduct the experiment, 12-well plates were used to place PAMs, which were then divided into five different groups. The first group served as the control group, which did not receive any treatment. The remaining four groups were treated with different stimulations—1 µg/mL LPS (Sigma-Aldrich, Steinheim, Germany), 20 ng/mL IL-4 (BIOTECH, INC, Alpharetta, GA, USA), 2 μg/mL of A3, or 2 μg/mL—of A4, respectively. After treating the PAMs with these stimulations for 24 h, the cells were collected and RNA extracted for further analysis using qPCR.

### 2.7. Gene Expression Profiling by Quantitative Real-Time Polymerase Chain Reaction

In this study, total RNA was extracted from the samples using Trizol reagent (Invitrogen, Waltham, MA, USA), a widely used chemical for isolating RNA from biological samples. The extraction process was carried out according to the protocol provided by the manufacturer. Once the RNA was extracted, it was reverse-transcribed into cDNA using the iScript cDNA synthesis kit (Bio-Rad, Hercules, CA, USA). This kit contains all the necessary enzymes and reagents for the reverse-transcription reaction, which converts the single-stranded RNA molecules into double-stranded cDNA molecules. The cDNA was then used as a template for quantitative real-time PCR, which is a widely used technique for measuring the relative expression levels of specific genes.

For the qPCR reactions, the KAPA SYBR FAST qPCR Master Mix Kit (KAPA Biosystem, Wilmington, DE, USA) was used, which contains all the necessary enzymes, buffers, and dNTPs for the PCR reaction. The reactions were performed using a Qiagen Rotor Gene Q Real-Time PCR machine (Qiagen, Germantown, MD, USA), which is a highly sensitive and accurate instrument for measuring the amplification of specific DNA sequences in real time. The specific primer sequences for the target genes are listed in the Appendix A. The amplification consisted of a 3 min step at 95 °C to denature the DNA, followed by 40 cycles of denaturation at 95 °C for 3 s and annealing at 60 °C for 20 s. The data were analyzed using the comparative Ct method, which is a widely used method for calculating the relative mRNA levels of specific genes. This method involves normalizing the target gene expression levels to those of a reference gene, using the equation 2^(−ΔΔCt)^, where ΔΔCt is the difference in Ct values between the target gene and the reference gene in the experimental and control samples.

### 2.8. Surface Protein Expressions by Flow Cytometry Analysis

PAMs were collected at a concentration of 10^6^/mL and cleansed with phosphate-buffered saline that contained 0.5% bovine serum albumin (BSA) (Sigma-Aldrich, Steinheim, Germany). They were then incubated with a specific monoclonal antibody that targetedproteins located on the surface of the PAM cells, which included SLA II^+^ (Bio-Rad), CD14^+^ (Invitrogen), CD80^+^ (Invitrogen), TLR4^+^ (Invitrogen), and CD163^+^ (Invitrogen). The cells were allowed to bind for 20 min while they were kept on ice and in the dark. Following this incubation period, flow cytometry was utilized to analyze the cells. The BD FACSDiva Software (BD Biosciences, CA, USA) and FlowJo Software (Tree Star, Inc., Ashland, OR, USA) were both used in the flow cytometry analysis.

### 2.9. Porcine Cytokine Assay

To analyze the phenotype of the immune reaction induced by the recombinant antigen A3, we cocultured 2 × 10^6^ PAMs with T-cell subsets. The coculture was performed using an indirect transwell coculture (Corning, Corning, NY 14831, USA) system in standard conditions (5% CO_2_; 37 °C) for 48 h.

Then, conditioned medium of PAMs and T-cells were collected for porcine cytokine assay. Cytokine concentrations were determined from a standard curve created with a reference preparation of commercial ELISA kit IL-10, IFN-γ (Thermo Fisher Scientific, Vienna, Austria), and IL-12 (R&D System, Abingdon, UK) according to the protocols provided by the manufacturer. The optical density A450 nm of each well was determined by EZ Read 400 Microplate Reader (Biochrom, Cambridge, UK).

### 2.10. Statistical Analysis

The data are expressed as means ± standard error of the mean where appropriate. A t-test with a 95% confidence limit and one-way ANOVA followed by Tukey’s test for multiple comparisons were used for statistical analysis. Data analysis was performed using Prism 8.0 software (GraphPad Software Inc., San Diego, CA, USA). Differences were considered statistically significant if the *p*-value was less than 0.05.

## 3. Results

### 3.1. A3 Directs Macrophages Polarization toward M1 Subtype and Downregulation of CD163 Expression

PAM cell lines that stably express CD163, an M2 macrophage marker, are highly susceptible to infection by both the PRRSV-1 Lelystad strain and PRRSV-2 VR-2332 strain [28]. Our study involved outlining a gating strategy that utilizes the characteristics of cell size and granularity, as measured by forward light scatter and side light scatter (FSC-A/SSC-A), to differentiate between positive and negative cells. Additionally, we excluded aggregated cells by using FSC-W/FSC-H and SSC-W/SSC-H measurements (Figure 1a). In our study, we found that in healthy PAMs, up to 99% of cells expressed high levels of CD163^+^ (Figure 1b). Interestingly, the PRRSV infection led to a significant decrease in the population of CD163^+^ and other markers of PAMs, suggesting that the cells were targeted for infection or killed by the virus. On the other hand, PAMs induced with the antigen A3 or A4 did not shrink the CD163^+^ population (Figure 1c). This finding is consistent with our earlier results on A1 [27], which indicated that our recombinant antigens did not significantly impact the expression of surface protein CD163.

However, we found that A3 was better able to decrease the mRNA expression of *CD163* in PAMs compared with that of A4 or the PRRSV infection (Figure 2a). The reduction in *CD163* expression in PAMs suggests a potential role of A3 in reducing the recycling of CD163 receptor that was susceptible to PRRSV infection. This result was in parallel to the previous result of A1 [27].

Furthermore, we also proved that A3 have the same ability as A1 in repolarization of PAMs toward M1, suggested by the upregulation of M1 macrophages genes marker (*TNF-α*, *IL-1β*, *IL-6*, and *IL-12*) and downregulation of M2 macrophages genes marker (*PPAR-γ* and *Arg-1*). In contrast, A4 did not reproduce the same effect on those genes, except for *PPAR* (Figure 2b).

### 3.2. A3 Stimulates the Expression of Toll like Receptors (TLRs) in PAMs

Toll-like receptors (TLRs) are a type of immune receptor that play a crucial role in detecting pathogen-associated molecular patterns (PAMPs) on various microbes as well as viruses, and initiating an innate immune response [29]. Our present study demonstrated that A1 was incapable of stimulating the expression of *TLRs* (Figure 3a). In contrast, we surprisingly found that *TLR3*, *7*, and *9* were significantly upregulated, up to 90-fold, in PAMs that were stimulated with A3, but not A4 (Figure 3b). This upregulation of *TLR*s in PAMs may devote to the activation of the innate and adaptive immune response and potentially enhance the effectiveness of vaccines against PRRSV infection.

Drawing on the findings of this study, we propose that A3 has the potential to activate immunity against PRRSV more effectively than A4. In order to test this hypothesis, further analysis of costimulation for T-cell differentiation status was conducted.

### 3.3. A3 Regulates Signaling Pathway-Related T-Cell Differentiation Status

In a previous study of ours, it was discovered that A1 was able to activate the Rap1 and C-type lectin receptor pathways, which were involved in the signaling mechanisms that influence T-cell differentiation status [27]. We tested the expression of certain genes that are related to signaling pathways unveiled from our previous transcriptomic analysis of PAMs induced by A1. The results indicated that A3, similarly to A1, upregulated genes that play a role in the regulation of the Rap1 signaling pathway (as depicted in Figure 4a), while there was no impact observed in cells with the induction of A4.

Additionally, we observed gene expression levels in our A3-induced PAMs (Figure 4b) according to the C-type lectin receptor (CLRs) pathway in A1-induced PAMs. Our data revealed that genes related to the C-type lectin receptor (CLR) pathway, such as *NF-kβ*, *p65/RELA*, *NFATC1*, and *IL-1β*, were significantly upregulated in the A3 group compared to the control or the PRRSV-infected PAMs (as shown in Figure 4b). Despite this, A4 only demonstrated a significant effect on the expression of *p56/RELA*. The representative genes regulating activation of the CLR pathway were confirmed by real time-PCR (Figure 4c). This was in accordance with the results showing that CLRs mediate T-cell receptor (TCR) signaling pathway activation, thus promoting proliferation, survival, and gene activation, especially of NF-κB [31].

### 3.4. TLRs in PAMs Stimulated by A3 Lead to Activation of Th1 Immune Response

Recent studies indicate that TLRs may be involved in the process of activating immune responses by activating APCs, which then activate CD4^+^ T-cells [21]. Most research has focused on the role of TLRs and the signaling pathways involved in regulating Th1-type immune responses [32,33]. According to these references, we then analyzed the gene expression and protein secretion that led to Th1 activation on both PAMs and T-cells in the coculture system. Our data demonstrated that the expression of *IL-12* was significantly upregulated in both cells (Figure 5a,b), but only PAMs secreted IL-12 and IFN-γ under A3 induction (Figure 5c,d). It is worth noting that the expression of the immunosuppressive cytokine IL-10 at both gene and protein levels was reduced by A3 induction in PAMs and T-cells compared to that of cells infected with PRRSV or induced by A4 (Figure 5). This result indicates that A3 can reverse the immunosuppressive effect of PRRSV infection.

## 4. Discussion

Developing a vaccine that provides protective immunity against PRRSV is a complex and ongoing challenge. While currently there are multiple available options for vaccines against PRRSV, their efficacy can be inconsistent. One of the limitations of these vaccines can arise from their ineffectiveness in targeting the TLRs of the pig’s immune system, which plays a key role in recognizing and responding to pathogens [34]. The protective immunity and distinct responses of many successful vaccines come from activating multiple TLRs [35]. TLR pathways play a role in vaccine responses by controlling the activation of adaptive immunity through various mechanisms, such as cross-priming of CD8^+^ T cells, inducing the expression of cytokines and costimulatory proteins, and reversing tolerance [16,36,37]. Additionally, TLR9 has been shown to be important in inducing the production of type I interferons (IFNs), particularly via plasmacytoid dendritic cells, during viral infection. TLR signaling and T-cell receptor pathways are known to be involved in host–vaccine interaction [38]. Therefore, understanding and enhancing the innate immune response, especially TLRs to viruses, is an important goal for vaccine development [37,39,40].

Our previous study found that recombinant PRRSV-2 antigen A1 stimulated the repolarization of M2 PAMs to M1, leading to a reduction in CD163 expression that provided broad protection against PRRSV-1 and PRRSV-2 strain infection. A1 also stimulated a Th1 response, activating the T-cell receptor signaling pathway in PAMs and causing the secretion of IFN-γ from T cells [27]. However, we did not observe any enhanced expression of *TLR*s. These receptors are known to play a role in the immune system, so their absence in our study was noteworthy.

In this study, we developed a new PRRSV-2 recombinant antigen in two different ways. We created ORF6L5, which is made up of the same components as A1, but has a different sequence of ORF5 and is without T-cell epitopes. The reason that we did not include T-cell epitopes was that we hypothesized that their presence might be the cause of the absence of *TLR*s in this antigen. A limitation of vaccines that primarily focus on T-cell epitopes is that viruses may have an increased chance of evading the immune response [41]. Viruses may use various strategies to evade the immune system, including changes in the amino acid sequences of epitopes, which can hinder recognition and processing by the immune system and potentially reduce the effectiveness of a vaccine [42]. We also altered the ORF5 sequences by eliminating the initial 20 amino acids in the GP5 complement of A3 to boost B-cell production. Nan et al. (2017) demonstrated that the main sequence of the B epitope was being located between amino acids 37 and 45 of PRRSV-GP5 [43]. The other recombinant protein is NLNsp10L11, which consists of ORF7 plus NSP 10 and NSP11. It is well established that nonstructural proteins (NSPs) are the first viral proteins to be produced in cells infected by PRRSV [44]. Following our previous study, we examined the expression of *CD163* in PAMs and the capacity for PAM repolarization by A3 and A4. Our findings showed that only A3 can reduce the mRNA expression of *CD163* in PAMs and cause M1 repolarization when compared to that infected with PRRSV at an MOI of 0.05 (Figure 2a. This was confirmed by the upregulation of M1 genes (*TNF-α*, *IL-1β*, *IL-6*, and *IL-12*) and the downregulation of M2 genes (*PPAR-γ* and *Arg-1*) in PAMs (Figure 2b).

TLRs play a key role in the immune system, so understanding whether A3 can stimulate their expression in PAMs could have important implications for the immune response to infection. However, instead of *TLR4* and *8*, we found that A3, but not A1 or A4, can trigger the expression of *TLR3*, *7*, and *9* in PAMs to the same extent as cells infected by PRRSV (Figure 3b). The result is in accordance with a study finding that the activation of *TLR7* had a significant impact on the deletion of *TLR8* in monocytes and dendritic cells [45]. TLRs 3, 7, 8, and 9 are intracellular TLRs that are located within cellular compartments such as endosomes and are able to detect nucleic acids, both DNA and RNA. When virus-derived pattern molecules are detected, these TLRs stimulate the production of antiviral genes, such as type I interferon, as a response [46,47]. Thus, TLR activation leads to the upregulation of costimulatory proteins, such as CD40, CD80 (B7.1), CD86 (B7.2), and CD70, on antigen-presenting cells, hence differentially activating the Th1 developmental pathway [21].

Since TLRs play a crucial role in the immune response, we then sought the signaling pathways involved in this response. According to our previous research, the Rap1 pathway is the most prominently activated pathway in PAMs stimulated by A1, and its activity is modulated by TCR activation [27]. Importantly, Rap1 signaling pathway enhances immune response against viruses by promoting activation and proliferation of T cells, enhancing their ability to kill virus-infected cells, and promoting the production of cytokines such as interferon-gamma [48]. In order to confirm the consistency of our previous findings, we collected some genes that are known to regulate the Rap1 pathway involving *PI3K*, *PKC-γ*, *PKD-1*, and *AFDN* from our transcriptomic analysis. Expectedly, apart from *PKD-1* and *AFDN*, the expression of *PI3K* and *PKC-γ* was significantly upregulated (Figure 4a). Our previous study had already reported the involvement of those genes in the Rap1 signaling pathway [27]. Meanwhile, some collected genes, such as *NF-kβ*, *BCL-3*, *p56*, *NFATC1*, and *IL-1β*, involved in C-type lectin receptor signaling were also detected from what we had measured by A1 induction from our previously unpublished data (Figure 4b,c). While CLR binds to a virus, it can trigger several signaling pathways, including the NF-κB, MAPK, and PI3K pathways, leading to the production of proinflammatory cytokines, such as interleukin-1β (IL-1β), IL-6, and tumor necrosis factor-α (TNF-α), as well as the activation of antiviral responses [49]. These findings suggest again that A3 has similarity to A1 in terms of effectiveness for a vaccine.

The immunological synapse between A1-induced PAMs and T cells can stimulate a Th1 response, including the activation of the T-cell receptor signaling pathway in PAMs and the release of IFN-γ. Similarly, our present results indicated that the expression of *IL-12* was increased in both PAMs and Th1 cells when they were in a coculture system stimulated with A3 (Figure 5a,b). However, only PAMs secreted IFN-γ and IL-12 (Figure 5c,d). The production of IL-12 by macrophages may be due to the activation of *TLR*s via A3 induction. Our findings align with the concept that TLRs have a major influence on the balance between the production of IL-12 and its related family members [50]. It has also been observed in mice that TLR9 ligands, such as CpG oligodeoxynucleotides, are potent inducers of IL-12p70—a member of the IL-12 family of cytokines—synthesis [51]. Importantly, it has been found that activating two or more TLRs can lead to higher levels of IL-12 production compared to a single TLR activation [52]. We found that after infection with PRRSV, the production of IL-10 was significantly increased in both PAMs and T cells. This suggests that PRRSV infection suppresses T-cell activation and may potentially induce T-regulatory (T-reg) cell activation, in line with prior results demonstrating that PRRSV infection significantly induced *IL-10* mRNA and protein expression and stimulated immunosuppression of host cells [53]. The emphasis here is that the IL-10 in both mRNA and protein expression was downregulated in PAMs induced by A3 (Figure 5a,c). On the other hand, A4 was found to upregulate the expression of *IL-10* mRNA in PAMs and T cells, indicating its potential ability to activate T-reg cells (Figure 5a,b), the same reaction to PRRSV infection. A potential explanation for how A4 may activate T-regs is the presence of nsp10. According to Chen et al. (2017), the nsp10 found in PRRSV can increase the promoter activity of CD83, a newly identified marker for activated T-reg cells via the signaling pathways of NF-κB and Sp1 [54]. Finally, we have identified a new recombinant protein derived from PRRSV-2 that has multiple functions for vaccine development: it can reduce *CD163* expression to avoid viral receptor-mediated entry, repolarize PAMs towards proinflammatory M1 macrophages, and stimulate *TLR*s as functional APCs to further induce Th1 immune response. The other advantage of A3 is related to its removal of 20 amino acids that potentially stimulate B-cell production. It is our hope that the A3 antigen will not only elicit an innate immune response but also stimulate the production of antibodies through the activation of B cells. The activation of antibody-secretion plasma cells is crucial for the development of humoral immunity against viral infections [55]. Apart from humoral immunity, the recombinant antigen A3 has proved its potential for initiating a linkage of innate and T-cell-mediated immunity to provide broad protection against PRRSV.

## 5. Conclusions

TLRs are important links between innate and adaptive immunity and help to mediate antiviral immune responses by recognizing virus infections, activating signaling pathways, and inducing the production of antiviral cytokines and chemokines. This is why TLRs are crucial during vaccination, as they play a key role in stimulating the immune response and promoting protection against pathogens. According to the results of our study, the recombinant PRRSV-2 antigen A3 not only repolarizes M1 PAMs and reduces *CD163* expression but also stimulates the activation of *TLR*s in PAMs. This *TLR* activation leads to subsequent CLR and TCR signaling activation for activating Th1 immune response. While the application of TLR signaling pathways to enhance vaccine effectiveness holds promise, there are still questions about efficacy, feasibility, cross-strain specificity, and safety that need to be further explored.

## Figures and Tables

**Figure 1 viruses-15-00775-f001:**
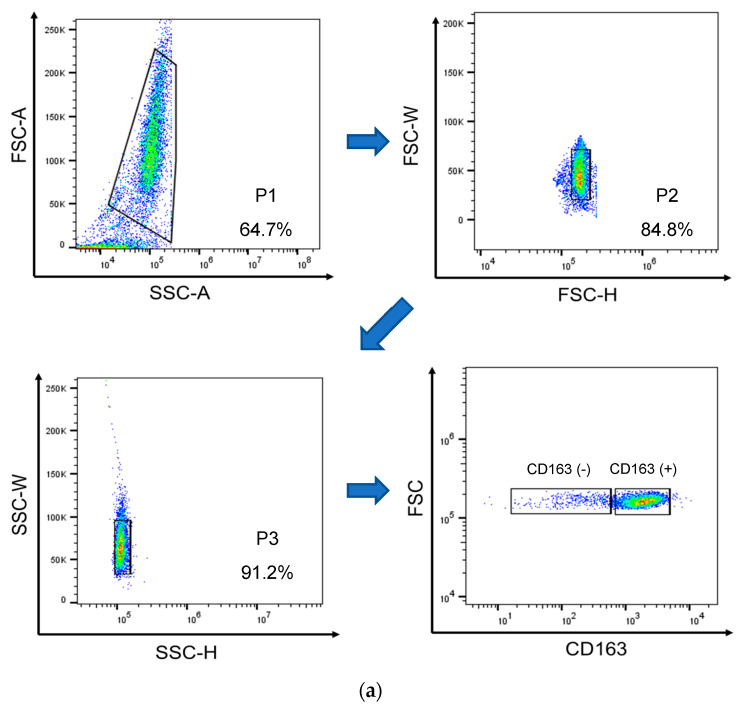
Fresh isolated PAMs expressed high levels of CD163 and susceptible for PRRSV entry. (**a**) Representative of gating strategy to distinguish between positive and negative cells. Dot plots (FSC-A vs. SSC-A) from a representative pig. The circles indicate living potential PAMs according to light-scatter properties (size and granularity). The arrows indicate the hierarchical sequences of analysis. (**b**) Expression of surface proteins SLA II^+^, CD14^+^, CD163^+^, CD80^+^, and TLR4^+^ on isolated PAMs by flow cytometry. High expression of CD163 was shown on fresh isolated PAMs. (**c**) Surface protein marker of PAMs after PRRSV infection or the recombinant antigen induction. PRRSV infection showed significant aberration in the percentage of CD163^+^ cells compared with normal cells. Cell percentage and mean fluorescence intensity (MFI) of SLA II^+^, CD14^+^, CD163^+^, CD80^+^, and TLR4^+^ were detected by flow cytometry. * *p* < 0.05 compared to control, θ *p* < 0.05 compared to 0.01 MOI, ψ < 0.05 compared to 0.05 MOI. Data presented as means ± SEM, calculated from 4 pigs.

**Figure 2 viruses-15-00775-f002:**
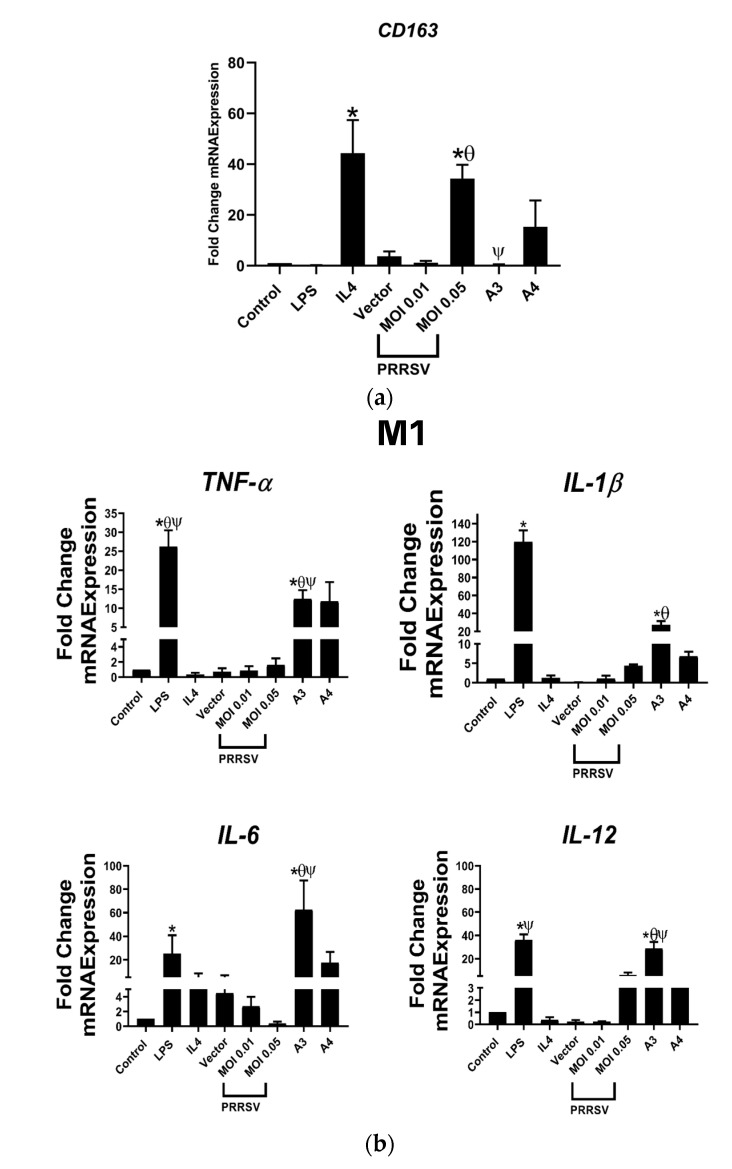
The antigen A3 decreased the expression of CD163 in PAMs and drove M1 macrophage polarization. (**a**) A3 decreased the expression of CD163 in PAMs. Untreated PAMs were used as control. The mRNA expression profile of (**b**) M1 and (**c**) M2 subtypes of PAMs measured by quantitative PCR (qPCR). A3 promoted upregulation of proinflammatory genes (M1 phenotypes) and downregulation of anti-inflammatory genes (M2 phenotypes). * *p* < 0.05 compared to control group, θ *p* < 0.05 compared to 0.01 MOI, ψ *p* < 0.05 compared to 0.05 MOI. Data presented as means ± SEM, calculated from 4 pigs.

**Figure 3 viruses-15-00775-f003:**
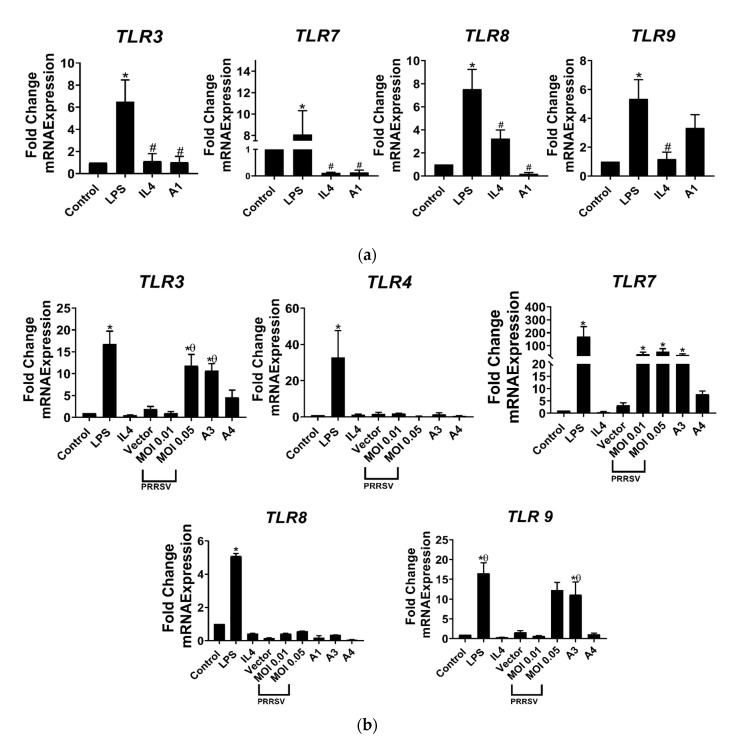
A3 induced the activation of *TLR*s, in parallel to PRRSV infection. (**a**) Data showed that A1 did not induce the activation of *TLR*s in PAMs. (**b**) A3 increased the expression of *TLR3*, *7*, and *9* in PAMs, but not *TLR4* and *8*. Untreated PAMs were used as control. * *p* < 0.05 compared to control group, # *p* < 0.05 compared to LPS group, θ *p* < 0.05 compared to 0.01 MOI. Data presented as means ± SEM, calculated from 4 pigs.

**Figure 4 viruses-15-00775-f004:**
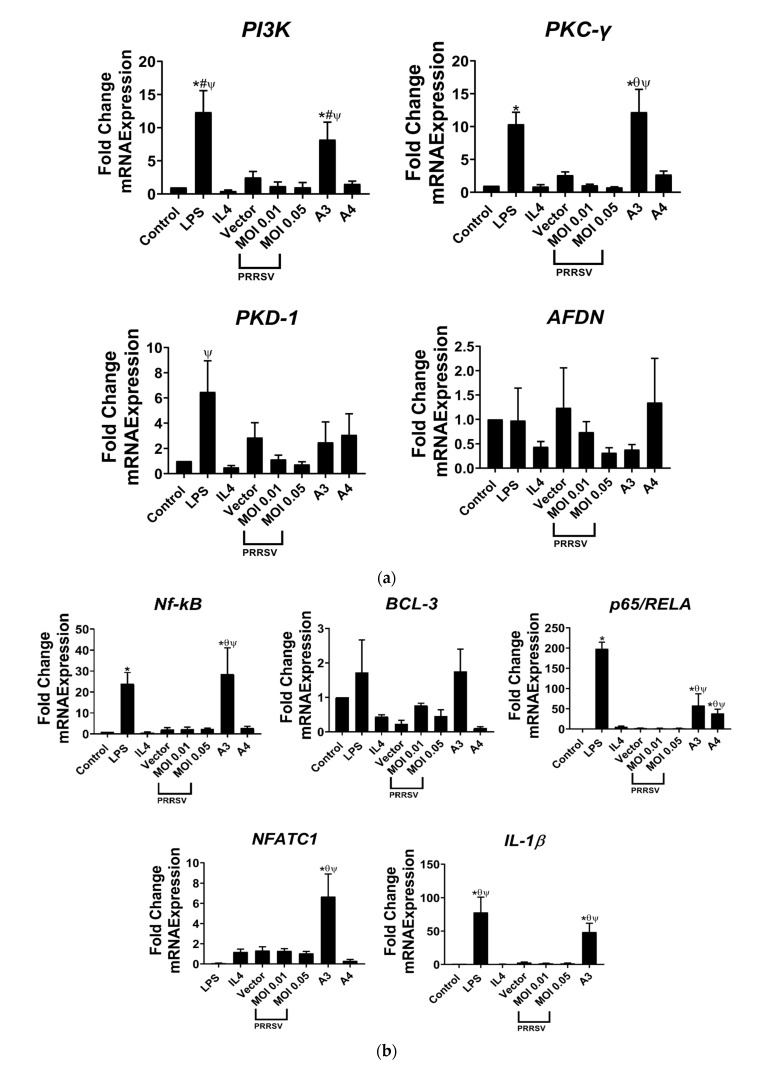
A3 drives signaling pathway-mediated T-cell differentiation status. (**a**) Representative genes regulate the Rap1 signaling pathway confirmed by real-time PCR. The Rap1 pathway is one of the key regulators of T-cell activation [30]. Those genes increased by A3 regulate the Rap1 signaling pathway. (**b**) Representative genes regulate the (**c**) C-type lectin receptor signaling pathway confirmed by real-time PCR. The representative genes regulating activation of the CLR pathway are represented by the red box. Untreated PAMs were used as control. * *p* < 0.05 compared to PAMs or CD4 group, # *p* < 0.05 compared to PAMs cocultured CD4 group (without any challenge or treatment), θ *p* < 0.05 compared to 0.01 MOI, ψ *p* < 0.05 compared to 0.05 MOI. Data presented as means ± SEM, calculated from 4 pigs.

**Figure 5 viruses-15-00775-f005:**
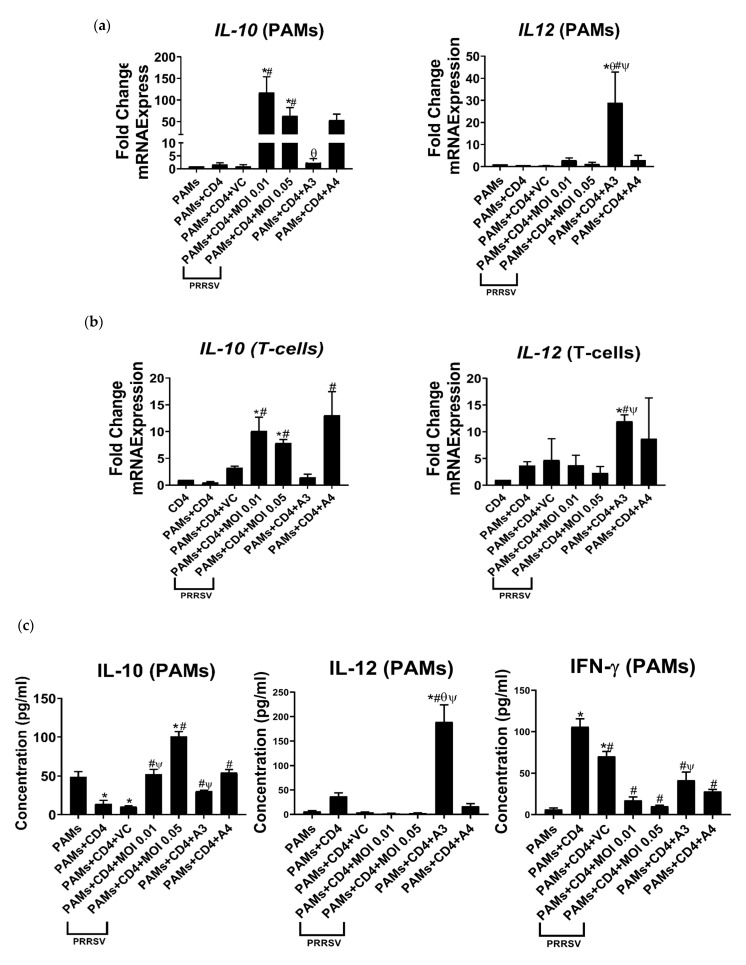
Th1 response modulated by A3-mediated *TLR* activation. Signature of (**a**) PAMs and (**b**) T-cell compartment in cocultured system by gene profile analysis. Signature of (**c**) PAMs and (**d**) T-cell compartment in cocultured system analyzed by ELISA to determine cytokine levels. Data presented as means ± SEM, calculated from 4 pigs. * *p* < 0.05 compared to PAM or CD4^+^ group, # *p* < 0.05 compared to PAM cocultured CD4^+^ group (without any challenge or treatment), θ *p* < 0.05 compared to 0.01 MOI, ψ *p* < 0.05 compared to 0.05 MOI.

## Data Availability

Data are contained within the article. Reported results can be found in Appendix A.

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
