# Peer review of "Toll-like Receptor-Mediated Immunomodulation of Th1-Type Response Stimulated by Recombinant Antigen of Type 2 Porcine Reproductive and Respiratory Syndrome Virus (PRRSV-2)"

_viruses, 2023, doi:10.3390/v15030775_

Round 1

Reviewer 1 Report

In this study, it has identified PRRSV-2 recombinant protein A3 that has multiple functions targeting on a vaccine development: it can reduce CD163 expression to avoid viral receptor-mediated entry, re-educate immunosuppressive M2 macrophages into pro-inflammatory M1 cells, and stimulate TLRs as a functional APCs to further induce Th1 immune response. This study provides a new mechanism for PRRSV vaccine design, and increases our understanding over the roles of M1/M2 polarization and TLRs activation in immune response. Several major and minor points are listed below for the authors to improve this manuscript during revision:

1.      It is suggested to describe the part of materials and methods for more details so as to make it easier for other researchers to repeat this experiment, for example, how to run the recombinant protein antigen constructs? Are the designed primer sequences derived from NCBI database (accession number?)., and so on.  

2.      Line 351-353, “According to these references, ……PAMs and T-cells in the co-culture system.” However, in the figure5, cytokine expression was detected in PAM and T cells, respectively. Therefore, it seems to be a contradiction here.

3.      There is almost no description of A4 in the discussion part, and it is suggested add a discussion/section describing the effects of A4 on the current result.

4.      Work was performed with a PRRSV-2 virus. Does the use of a PRRSV-1 virus produce the same outcome?

Reviewer 2 Report

The article by Wahyuningtyas et al. showed for the first time that a recombinant A3 antigen of the PRRSV-2 can potentially be used as a therapeutic. A3 stimulation in PAMs can trigger TLR activation and Rap-1 signaling leading to induction of innate immune responses, preferentially polarize macrophages into M1 subtype, and potentially enhance Th1 differentiation. The article is well organized and detailed, especially the methods. As a reviewer, I believe there is a load of data in the figures which are not sufficiently described in the text of the results section. Discussing the figures extensively in the results sections will be informative and will provide unbiased information to the readers. Other than that, there are language and grammatical errors, which can be improved using a proof-reading software. The authors have done significant work which will provide valuable information to the PRRSV field. Here are some comments according to me:

·      Introduction section: Necessary details about PRRSV-2 like genome type, structure, prevalence, how are they transmitted, can be added. Especially, because ORFs and NSPs are mentioned in the methods section, it may be beneficial to describe the viral genomic structure and/or add a figure for it.

·      The title mentions PRRSV-2 while starting from Introduction till the end of the article, “PRRSV” is mentioned in place of “PRRSV-2”. It will be beneficial to include that PRRSV-2 will be referred as “PRRSV” in this article, or kindly replace “PRRSV” to “PRRSV-2” wherever necessary.

·      Table 1: the gene names should be written in prover convention.

·      There is no mention of Figure 1b and 2a in the text.

·      In the figures with bar graphs, the bars are randomly compared with each other. Either all the bars should be compared separately with control, 0.01 MOI and 0.05 MOI, or just choose to compare all data with 0.05 MOI to simplify. At the current form, the data becomes unclear and biased. Statistical analysis should be performed and displayed on all data equally.

·      Kindly expand the result sections, as the figures have a lot of data/information in them. For example, nothing about A4 is mentioned in the text for Figure 1 and 2 although it looks like it has similar trends like A3 in most cases.

·      Use gene naming conventions for mRNA names. Also, keep naming consistent throughout the article (for example, write "NF-kB” instead of “Nf-kB")

·      Discussion section: Add briefly about what is known about the role of Rap-1 and C-type lectin receptor pathways in PRRSV or related virus infection.

Line 22: Please replace “0,01” and “0,05” to “0.01” and “0.05”, respectively.

Line 32-33: “provided a more potential in protecting” can be replaced with “provided a better protection against”.

Line 37: Keywords should be expanded words and not abbreviations.

Line 48-53: These sentences about TLR/Th1 are too generalized, while introduction of this article should be PRRSV specific. Hence, instead of these lines, this space may be used to describe why Th1 (and not Th2) response was focused on by the authors with regard to PRRSV. TLR specific information is described in rest of the introduction section, so that can be skipped here.

Line 64: “responses5” – maybe 5 was meant to be reference?

Line 157: Typing error in “106”

Line 160: When denoting marker-positive cell populations, the paper uses “+” in superscript in some places and without superscript in others. Please be consistent.

Line 163: Please provide gating scheme in a supplemental figure that clearly show specific marker-positive populations of PAM from marker-negative populations. Was a live dead stain used? Was doublet discrimination performed? Please indicate in the gating scheme.

Line 186-191: This section can go in the introduction. Alternatively, the aim of these experiments or hypothesis can be used to start the paragraph.

Figure 1a: Spelling error in x-axis.

Line 221: “To our findings” can be replaced with “We found”.

Figure 2: Why is MOI 0.05 compare with itself (according to the sign on top of the bar)?

Line 281: Kindly remove the “unpublished” word

Figure 3: There is no mention of # sign in the legend/caption.

Line 288: Add reference.

Line 355-358: This sentence needs to be reworded.

Line 472: Add reference.

Line 474: Add references of role/involvement of these genes in Rap1 pathway.

Line 503-509: References need to be added.
